# Impact of p85α Alterations in Cancer

**DOI:** 10.3390/biom9010029

**Published:** 2019-01-15

**Authors:** Jeremy D. S. Marshall, Dielle E. Whitecross, Paul Mellor, Deborah H. Anderson

**Affiliations:** 1Cancer Research Group, University of Saskatchewan, 107 Wiggins Road, Saskatoon, SK S7N 5E5, Canada; jem826@mail.usask.ca (J.D.S.M.); dielle.whitecross@usask.ca (D.E.W.); pam187@mail.usask.ca (P.M.); 2Department of Biochemistry, University of Saskatchewan, 107 Wiggins Road, Saskatoon, SK S7N 5E5, Canada; 3Cancer Research, Saskatchewan Cancer Agency, 107 Wiggins Road, Saskatoon, SK S7N 5E5, Canada

**Keywords:** PI3K (phosphatidylinositol 3-kinase), cancer, mutations, p85α subunit, PTEN (phosphatase and tensin homologue deleted on chromosome 10), Rab5

## Abstract

The phosphatidylinositol 3-kinase (PI3K) pathway plays a central role in the regulation of cell signaling, proliferation, survival, migration and vesicle trafficking in normal cells and is frequently deregulated in many cancers. The p85α protein is the most characterized regulatory subunit of the class IA PI3Ks, best known for its regulation of the p110-PI3K catalytic subunit. In this review, we will discuss the impact of p85α mutations or alterations in expression levels on the proteins p85α is known to bind and regulate. We will focus on alterations within the N-terminal half of p85α that primarily regulate Rab5 and some members of the Rho-family of GTPases, as well as those that regulate PTEN (phosphatase and tensin homologue deleted on chromosome 10), the enzyme that directly counteracts PI3K signaling. We highlight recent data, mapping the interaction surfaces of the PTEN–p85α breakpoint cluster region homology (BH) domain, which sheds new light on key residues in both proteins. As a multifunctional protein that binds and regulates many different proteins, p85α mutations at different sites have different impacts in cancer and would necessarily require distinct treatment strategies to be effective.

## 1. Introduction

Class IA phosphatidylinositol 3-kinase (PI3Ks) consist of an 85 kDa regulatory subunit (p85), partnered with a 110 kDa catalytic subunit (p110). The p85 isoforms are encoded by three genes, *PIK3R1* (p85α, p50α, p55α), *PIK3R2* (p85α) and *PIK3R3* (p55α). All the isoforms contain a pair of SH2 (Src homology 2) domains (nSH2, cSH2), flanking an interSH2 (iSH2) domain. These C-terminal domains are responsible for mediating binding to upstream tyrosine phosphorylated sites on receptors or adapter proteins, and to the p110-PI3K protein. In addition to the C-terminal domains, the larger p85 isoforms contain N-terminal regions that include: an SH3 (Src homology 3) domain and a breakpoint cluster region homology (BH) domain that has GTPase activating protein (GAP) activity (Figure 1). These regions bind to small GTPases and the lipid phosphatase PTEN (phosphatase and tensin homologue deleted on chromosome 10).

The p85α protein has a well-characterized role in binding, stabilizing and regulating the activity of the p110α catalytic subunit of PI3K [1]. Class IA PI3K is typically activated by upstream tyrosine kinases, either by activated receptor tyrosine kinases like the epidermal growth factor (EGF) receptor, or by cytoplasmic tyrosine kinases such as Src-family kinases [2,3]. Tyrosine phosphorylation of these kinases, or intermediate adapter proteins, creates binding sites for the SH2 domains within p85α, which simultaneously relocalizes the pre-existing p85α–p110α complexes to these intracellular locations (typically at the plasma membrane), and relieves the catalytic repression of p85α towards p110α [4,5]. The membrane localized and activated p85α–p110α-PI3K phosphorylates the 3′-position on phosphatidylinositol 4,5-bisphosphate (PI(4,5)P_2_) to generate phosphatidylinositol 3,4,5-trisphosphate (PI(3,4,5)P_3_), a key signaling lipid.

PI(3,4,5)P_3_ formation recruits pleckstrin homology domain containing proteins, including PDK1 (phosphoinositide-dependent protein kinase-1) and Akt to the membrane, facilitating the phosphorylation and activation of downstream Akt signaling [3,6,7], important for cell growth, cell cycle entry and progression, protein translation, and cell survival [7,8,9]. Activated receptor tyrosine kinases are internalized by clathrin-mediated endocytosis, a process involving small GTPases such as Rab5 [10,11]. After endocytosis, receptors are either deactivated and recycled back to the plasma membrane, or sorted for degradation in the lysosome through Rab-mediated trafficking [10,11]. Both pathways inactivate upstream receptor signaling, allowing the inactivation of the PI3K/Akt signaling pathway. It has been established that p85α binds to and downregulates several small GTPases, including Rab5 [12]. Mutations in p85α that inactivate its Rab5 regulatory activity, give rise to increased levels of activated receptors, resulting in cell transformation [13].

The activity of the PI3K pathway is counteracted by the lipid phosphatase PTEN which dephosphorylates PI(3,4,5)P_3_ back to PI(4,5)P_2_, preventing sustained downstream activation of the Akt pathway [14,15,16]. The p85α protein is bound to p110α with a strong interaction and is also required for p110α stability [1,17]. When p85α levels are in excess of p110α, the p85α protein homodimerizes [18] and binds directly to PTEN in response to growth factor stimulation, an interaction that improves PTEN stability and stimulates the catalytic activity of the PTEN protein [18,19,20]. Thus, p85α is uniquely positioned to both positively and negatively regulate the PI3K pathway through its interactions with protein partners including p110α, Rab5 and PTEN [11].

The PI3K pathway is dysregulated in a large number of cancers, including those of the endometrium, urothelial tract, breast, prostate, colon and brain (Figure 2; and reviewed in [5,8,11,21,22,23,24,25,26,27,28]. Oncogenic mutations can occur in various components of the pathway, including receptor tyrosine kinases and activating mutations in the p110α catalytic subunit or regulatory subunit p85α that leads to constitutive PI3K/Akt pathway activation. Loss of function mutations or deletions of PTEN, and/or the p85α protein can similarly result in sustained PI3K pathway activation. Various drugs targeting PI3K and other constitutively activated kinases within the pathway have been developed to treat dysregulation of the PI3K pathway [2,22,25,29]. Increasing evidence suggests that the effectiveness of these drugs may, in part, be dependent upon the specific mutations present in specific target proteins, and the genetic context in which that mutation occurs. A better understanding of how p85α mutations impact its interactions and influence its regulatory functions should provide a better understanding of which cancer-associated mutations lead to dysregulation of the PI3K pathway and could help identify the most effective pathway components to target with inhibitors.

## 2. p110α–p85α Interactions and Alterations

The p110αprotein consists of an adapter binding domain (ABD), a Ras binding domain (RBD), a C2 domain, a helical domain and a kinase domain. The C2 domain of p110α interacts with the iSH2 domain of p85α and the helical domain of p110α binds to the p85α nSH2 domain [5]. p110α is constitutively associated with p85α in a pre-formed complex localized to the cytosol, until growth factor stimulation relocates it to activated receptors at the plasma membrane.

According to the cBioPortal database [30,31], missense mutations and other alterations in the gene encoding p110α, *PIK3CA*, are observed in 11% of all cancers, with considerably higher frequencies in many cancer types including: endometrial (52%, 304/586), vaginal (50%, 2/4), penile (43%, 3/7), cervical (39%, 18/46) and breast (38%, 732/1923) (Figure 2a, Figure 3a and Figure 4a) [30,31]. Other p110 isoforms (p110α, p110α, p110α) are not typically mutated in tumors, but are instead overexpressed, which can be sufficient to mediate oncogenic transformation [32,33,34,35,36]. Gene amplifications of *PIK3CA* are also common in some cancer types and can contribute to an aggressive phenotype [37] (Figure 4a). Most *PIK3CA* mutations are gain-of-function mutations that activate p110α, resulting in increased downstream Akt signaling, promoting aberrant cell survival and tumorigenesis. The majority of these mutations occur in hotspots in either the kinase or helical domains and are oncogenic mutations (Figure 3a) [38,39,40,41]. The cancer-associated H1047R mutation is located within the p110α kinase domain (Figure 3a) and results in catalytic activation that still requires p85α binding, but is independent of Ras [27,42,43]. The H1047R mutation also seems to increase the attachment to the membrane [44]. In contrast, the cancer-associated E542K and E545K mutations are located within the helical domain (Figure 3a) and these p110α mutants still require binding to Ras [27]. These mutations activate p110α by disrupting the inhibitory interactions with the p85α nSH2 domain (residues K379 and R340) [42,43].

N345K is another frequent C2 domain p110α mutation that disrupts interactions with p85α residues D560 and N564. Interestingly, the corresponding mutations, D560G and N564D, have also been identified in the *PIK3R1* gene encoding p85α. The functional effects of each of these mutations is to remove the inhibitory influence of p85α on p110α, resulting in p110α activation [45,46].

Alterations in the *PIK3R1* gene are less common than *PIK3CA* mutations with an average of 3% across all cancer types. *PIK3R1* alterations are found at high frequency in several cancer types, including: endometrial (29%, 172/586), glioblastoma (7%, 41/593), colorectal (6%, 38/594) and prostate (6%, 4/70) (Figure 2b) [30,31]. The most common p85α point mutations are located within the nSH2 domain (G376R and K379N/E) and iSH2 domain (D560G, N564D and K567E) (Figure 3b). Mutations in the nSH2 domain of p85α, such as the K379E mutation, may have the same effect as mutations such as the hotspot E542K and E545K mutations in the helical domain of p110α [5,47,48]. They are gain-of-function mutations as a result of the loss of p110α inhibition and enhanced PI3K signaling [38,39,46]. The effect of p85α mutations within the C-terminal half of the protein responsible for p110α stabilization, relocalization and regulation are more thoroughly discussed in an excellent recent review article [22].

## 3. Small GTPases–p85α Interactions and Alterations

Rho GTPases are signaling G protein members of the Ras superfamily and include Rho, Rac and Cdc42. These small GTPases function to regulate the organization of the actin cytoskeleton and focal adhesions, which are important factors influencing cell migration and invasion [49]. Small GTPases have two functional conformations: the inactive GDP-bound conformation, and the active GTP-bound conformation which is able to bind multiple protein effectors. Both Cdc42 and Rac, but not RhoA, have been shown to function as upstream regulators of PI3K. Studies by Zheng et al. identified direct binding of active Cdc42 and Rac to the BH domain of p85α, stimulating PI3K activity [50]. This increase in PI3K activity appears to be situation dependent and may only account for a small proportion (<1%) of total PI3K activity [51,52]. In fact, the final goal of this interaction may not be to increase PI3K activity but rather to target the small GTPases to specific sites. The decrease in actin stress fibers and the development of filopodium by activated Cdc42 has been shown to be dependent on Cdc42 binding to the p85αregulatory subunit, but not PI3K activity, following stimulation of the platelet-derived growth factor receptor α (PDGFRB) [53]. It is thought that the change in the subcellular location of p85α following PDGFRB stimulation enables recruitment of Cdc42, which is then activated through the actions of guanine nucleotide exchange factors (GEFs) at that site, bringing about cytoskeletal changes [53].

Alternatively, the presence of activated Rac/Cdc42 at the leading edge of a cell may recruit additional PI3Ks leading to the production of PI(3,4,5)P_3_ and the subsequent sustained activation of Rac-GEFs, including Vav1 and Sos1, both of which require PI(3,4,5)P_3_ for their activity [54,55]. This creates a positive feedback loop due to activation of PI(3,4,5)P_3_ regulated GEFs at the cell membrane.

The Rho GTPases have been implicated in performing roles in many of the steps of cancer ranging from tumor initiation to metastasis, which have previously been summarized in a couple of reviews [56,57]. Rho GTPases are overexpressed in a number of cancers, and alterations in both upstream and downstream regulators can also contribute to tumorigenic properties [58]. More recently some tumors have been identified that contain a low frequency of activating mutations in Rac and Cdc42 in melanoma, breast and neck cancer [59,60,61,62]. These are recent discoveries and more work is required to determine the impact of these mutations on cancer cell properties.

Binding to the BH domain of p85α can also regulate several small GTPases, including Rab4 and Rab5 [12,50,52]. Rab5 is involved in the uptake and trafficking of receptors, integrins, and other cell surface proteins into the cytoplasm via the early endosome [63,64,65,66]. As other GTPases, Rab5 cycles between an active GTP-bound and inactive GDP-bound state. For Rab5, the cycling between these states controls early endosomal trafficking events by regulating protein recruitment to vesicles necessary for tethering and fusion of the endosomal membranes, lipid modifications to the membrane composition, and movement [67,68,69]. While Rab-family proteins have intrinsic GTPase activity, allowing for a level of self-regulation, they are also regulated by GAPs which significantly accelerate the rate of hydrolysis of GTP to GDP [68,70,71]. The BH domain of p85α is one such Rab5-GAP though its activity is quite weak [12,72]. The role of p85α-mediated regulation of Rab5 function was also demonstrated during the processes of bacterial invasion [73] and cell motility [66].

Cell migration is a complex process involving the formation of actin-mediated protrusions, the transient formation of integrin-mediated focal adhesions to the extracellular matrix at the leading edge, and actomyocin-mediated contraction with adhesion disassembly at the rear of the cell, to allow the cell to slide forward [74]. In vivo, cells also require extracellular matrix degradation to provide space for the cells to move [75]. Rab5 facilitates protein complex assembly in cell migration and invasion to help degrade the extracellular matrix [75,76], remodel actin structures [77,78], and regulate focal adhesion turnover to allow cells to attach at the front and to detach at the back [75,76,79]. Specifically, Rab5 overexpression has been shown to promote matrix metalloprotease release and activation, contributing to extracellular matrix degradation [75,76]. Additionally, Rab5-GTP levels are increased at the leading edge of migrating tumor cells where it interacts with RIN2 and recruits Tiam1, a Rac guanine nucleotide exchange factor that activates Rac [80]. This leads to actin polymerization, stimulating the formation of lamellipodia and membrane ruffles at the leading edge of a migrating cell. Rab5 also associates with α1 integrins, focal adhesion kinase (FAK), vincillin and paxillin and plays an essential role in focal adhesion disassembly [76,79,81]. Rab5 increases the rate of internalization and recycling of α1 integrins as they move from the back of a migrating cell to the leading edge [79,81,82,83]. Thus, Rab5 has been described as the master regulation of endocytosis and its overexpression has been suggested to contribute to more aggressive and metastatic cancer [75].

Rab5 has been found to be upregulated in many cancers including colorectal cancer [84], breast cancer [75,85], cervical cancer [86], liver cancer [82], lung cancer [87] and pancreatic cancer [88,89]. Overexpression of *RAB5A* is an indicator of poor prognosis in hepatocellular carcinoma, pancreatic cancer and estrogen receptor positive breast cancer [86,90,91,92]. These studies demonstrated that migration and invasion of cancer cells are directly affected by Rab5 expression levels. The increased Rab5 levels in these cancer cells may overwhelm Rab5 regulatory proteins such as GAPs—allowing for an increase in active Rab5 as compared to normal cells and disruption of the temporal regulation of endosomal traffic. Indeed, the re-expression or overexpression of the dominant negative mutant Rab5-S34N does not mimic the overexpression of wild type Rab5, indicating that active Rab5-GTP is responsible for these phenotypic effects [75,83].

All of these activities require active Rab5, suggesting that mutation or changes in expression of GAPs and GEFs would be able to influence cell migration through their regulation of Rab5 [93,94]. As a GAP protein that stimulates the downregulation of Rab5 to the inactive Rab5-GDP form, p85α is also involved in regulating Rab5-mediated processes, including cell migration. In support of this, caveolin-1 was shown to sequester p85α [75], which promoted Rac activation, Rab5-dependent endocytosis and migration of cancer cells [95]. Sequestration of p85α would block p85α-mediated Rab5 inactivation and increase cell migration. This suggests that inhibitors which block the caveolin-1–p85α interaction could function as new therapies to reduce cancer metastases.

There is another example where sequestration of p85α has been shown to drive increased Rab5-dependent migration. Caspase 8 is a cysteine protease involved in apoptotic signaling that also associates with early endosomes [96,97,98] and plays a role in cell motility [99,100,101,102]. Phosphorylation of caspase 8 was found to enhance cell motility by binding p85α and sequestering it away from Rab5, resulting in increased Rab5-GTP [103,104]. In a subsequent study, integrin engagement induced Rab5 activation and association with α1 integrin complexes [81]. Caspase 8 expression was also shown to promote Rab5-mediated α1 integrin internalization and recycling, increasing cell migration, which was independent of caspase enzymatic activity [81]. Thus, sequestration of p85α by caspase 8 enhanced Rab5-GTP-dependent processes, presumably by keeping p85α-encoded GAP activity away from Rab5.

As mutations in p85αthat compromise its GAP activity (e.g., R274A, R151A, E137K) have been shown to result in increased Rab5-GTP levels and oncogenic cell properties [76,105], similar results would be expected for mutations in either p85α (e.g., L191D, V263D) or Rab5 that abrogate binding [105]. Oncogenicity resulted from the altered kinetics of Rab5- and Rab4-mediated PDGFRB internalization, dephosphorylation and recycling that gave rise to sustained downstream signaling [13]. Moreover, the correlation between increased *RAB5A* expression [75,82,84,85,86,87,88,89] and decreased *PIK3R1* expression in cancer [106,107] warrants further investigation, as a possible feature of oncogenic adaptation where metastasis via Rab5-mediated cell migration progresses unchecked due to loss of regulation by p85α.

## 4. PTEN–p85α Interactions and Alterations

PTEN is composed of five domains: a plasma membrane binding domain that binds PI4,5P_2_ (PMB), a dual lipid/protein phosphatase domain (PASE), a C2 domain (C2), a Ser/Thr rich regulatory domain (REG), and a PDZ binding domain (PDZB). In the absence of growth factor stimulation, PTEN is constitutively phosphorylated on residues S370, S380, T382, T383, and S385 within the PTEN regulatory domain by casein kinase 2, GSK3β (glycogen synthase kinase 3β), PICT-1 (protein interacting with carboxyl terminus 1), and ROCK (Rho-associated coiled-coil-containing protein kinase 1) kinases [108]. These phosphorylated residues are thought to interact with basic residues found on a surface of the phosphatase (R161, K163, K164) and C2 (K260, K263, K266, K267, K269) domains to form a closed conformation [109]. The closed conformation of PTEN keeps it localized in the cytosol and increases the stability of the protein [15]. Stimulation from upstream activating signals, such as EGF stimulation, triggers the dephosphorylation of PTEN which assumes an open conformation to allow for interactions with lipids at the plasma membrane [15,16,109,110,111]. PTEN homo-dimerization has also been shown to play a role in its lipid phosphatase activity [112].

The EGF-stimulation has been shown to induce p85α association with the unphosphoryated form of PTEN, and this PTEN–p85α interaction stimulates PTEN lipid phosphatase activity [19,20]. The binding of p85α to PTEN stimulates PTEN lipid phosphatase activity and is mediated by both the SH3 and BH domains of p85α, and the phosphatase and C2 domains of PTEN [18,20,113]. Truncation mutations in p85α (e.g., E160*), give rise to a protein that lacks the BH domain and other C-terminal domains. These function as dominant negative mutants by binding to the wild type full-length p85α forming a non-productive dimer that sequesters it from PTEN [114]. As a result, p85α does not stimulate PTEN activity and leads to reduced levels of PTEN protein [114].

The ability to form p85α homodimers is critical for their interaction with the PTEN protein and is necessary for PTEN stability [18]. Homodimerization of p85αhas been suggested to be mediated by reciprocal interactions between the SH3 domain of one monomer with the PR1 region on a separate p85α monomer [115], together with interactions between the two cSH2 domains [116] and may also involve BH–BH domain interactions [18,115,117]. Mutations of residues within the p85α SH3 domain (D21A and W55A or E19A and E20A) and BH domain (I133N, I177N, or M176A) resulted in decreased p85α homodimerization and binding to PTEN [18]. These mutant proteins caused an increase to PTEN ubiquitination and a decrease in total PTEN protein levels. The p85α homodimers were found to compete with the E3 ubiquitin protein-ligase WWP2, and the disruptions to p85α homodimerization lead to increased proteosomal degradation of PTEN from increased ubiquitination [18]. Thus, other mutations and/or truncations in p85α that block its homodimerization, would be expected to cause similar reductions in PTEN stability.

Disruption of the PTEN–p85α interaction has a considerable negative impact on PTEN levels resulting in sustained activation of the PI3K/Akt pathway as noted above. To identify key residues important for the PTEN–p85α interaction our lab recently carried out extensive deletion and mutagenesis analyses [113]. Our initial experiments demonstrated that PTEN mutations K342E and K344E blocked p85α binding, and p85α mutations D168R, E212R, E218R, K224E + K225E, R228E, H234D, Q241D and K249E disrupted PTEN binding. We subsequently carried out a pair of docking analyses, using the previously solved structure of PTEN (PDB ID# 6D81; [110]) with either of the solved structures of the BH domain of human p85α (PDB ID# 1PBW; [117]) or bovine p85α (PDB ID# 6D81; [105]). One model emerged as the most consistent with the available binding data. This model shows extensive interactions between the PTEN PASE domain (R84, Q87, Y88, E91, E99) and the p85α BH domain (Q214, E212, Q221, K224, K225), as well as the PTEN C2 domain (R189, R190, Q219, C250, D252) and the p85α BH domain (R228, H234, W237, K245, E297). Some of these interactions are shown in Figure 5. Subsequent experiments testing multiple mutations in the p85α BH domain, or PTEN-E91R mutation lend further support to this model [113].

The PTEN–p85α interaction model is consistent with other available data for both PTEN and p85α, in that it can accommodate PTEN plasma membrane association [111], PTEN homodimerization [112], p85α BH domain homodimerization [117] and p85α BH domain Rab5/GTPase binding [105,113]. The model also has space for other PTEN and p85α domains not present in the protein fragments modeled. A previous study showed a modest reduction (~35% decrease) in PTEN binding for a triple mutant p85α-I127A + I133A + E137A [18]. The location of these residues within our model suggests they could associate with the PTEN C-terminal regulatory region (not in the model), which is positioned along that p85αBH domain surface (Figure 5). This newly generated model for the PTEN–p85α BH domain interface can be used to predict the consequences of cancer-associated mutations in key residues expected to disrupt binding and destabilize PTEN. Based on this model, there are cancer-associated mutations in PTEN that would be predicted to disrupt binding to p85α, including: D84G, Q87P, Y88C/H/N/S, E91Q, E99K, R189K, K342N and K344R [113]. Similarly, there are cancer-associated mutations in p85α that would be predicted to disrupt binding to PTEN, including: Q214L, W237L and K245T, according to the COSMIC [61] and cBioPortal [30,31] databases.

*PTEN* loss or mutation is a frequent event in many cancer types, particularly in endometrial (62%, 364/586), glioblastoma (32%, 187/593) and prostate (22%, 124/564) (Figure 2c) [30,31]. The most common hotspot mutation (R130Q/G/L) (Figure 3c) renders PTEN inactive and can exert dominant negative effects by constraining the phosphatase activity of the co-expressed wild type PTEN through dimerization [112]. Since PTEN is stabilized by p85α binding, mutations in PTEN and/or p85α that disrupt this interaction would have similar impacts, to reduce PTEN levels and PTEN activity, resulting in an upregulation of the Akt pathway.

Alterations in *PTEN* and *PIK3R1* also include a high frequency of deletions (Figure 4b,c) and truncating mutations (Figure 3b,c), resulting in either reduced expression or dominant negative effects as noted above. *PIK3R1* expression is reduced in some cancer types as compared to that in their corresponding normal tissue, including cancers of the ovary, prostate, breast, lung, liver and kidney [106,118]. Reductions in *PIK3R1* is associated with poor prognosis in breast and lung cancer [106,107], likely due to its important role in PTEN stabilization. In animal models of cancer, liver-specific knockout of *PIK3R1* increased the activation of the PI3K pathway and caused tumorigenesis [118]. They also showed a corresponding decrease in PTEN levels, consistent with the loss of p85α-mediated PTEN stabilization. Decreased expression of p85α through heterozygous deletion of *PIK3R1* was sufficient to increase the frequency of intestinal polyps in the context of a PTEN heterozygous background from 30 to 60%, when compared to PTEN heterozygous mice [119]. Thus, decreased levels of p85α can contribute to tumorigenesis.

## 5. Characterizing Variants of Unknown Significance

The vast majority of cancer-associated alterations are of low frequency and remain variants of unknown significance (Figure 4). This represents a major obstacle in designing effective treatment strategies, since the biological impact of the numerous different mutations are not known, and variants of unknown significance are typically present in multiple proteins within a single tumor. OncoKB is a database that annotates the effects of mutations and treatment implications based on information gathered from publications, and makes them available through a public web resource (http://oncokb.org) [120]. This information has also been included in cBioPortal for Cancer Genomics, as a way to provide some of the mutational consequences to oncologists and researchers. Another strategy utilized a computational analysis of 25,000 cancers, including >10,000 from patients with advanced and/or previously treated disease [121]. The scale of the data analysis allowed nearly 1200 statistically significant hotspot mutations to be identified, with 80% occurring at frequencies of less than 1 in 1000. A small subset of patients derived clinical benefit when they received treatments based on these newly identified hotspot mutations [121]. These results suggest that as more genomic and clinical data becomes publicly available for analysis, e.g., through initiative such as the AACR Project GENIE [122], more mutational hotspots will be identified and the number of patients to benefit from available treatments will increase.

The discovery of cancer driver mutations does not effectively predict therapeutic benefit in the absence of additional information. Computational analyses have been used in conjunction with tools such as protein structure data to define colocalized mutations within the folded protein [122]. Network analysis, machine learning and database mining strategies are also being applied to factor in clinical information and other data [122]. Understanding the impact of multiple combinations of mutations and other genetic alterations on cancer biology and treatment sensitivity further complicates the field.

Systematic functional characterization of cancer-associated mutations is a huge challenge, but is essential for implementing precision oncology. A recent report describes a moderate-throughput functional genomics platform and uses it to annotate >1000 cancer variants of unknown significance [123]. This study focused on clinically actionable genes where current targeted therapies were available (e.g., *EGFR*, *PIK3CA*, *BRAF* and *ERBB2*), as well as potentially druggable genes (e.g., *PTEN*, *ALK*, *PDGFRA* and *FGFR2*). Two cell lines were used for each target to determine the growth factor-independent cell viability when the mutant was expressed, as compared to the wild type control. A functional proteomic analysis was also carried out on a subset of these cell lines using a reverse phase protein array (RPPA) to quantify target protein levels and specific post-translational modifications [123]. Results could be clustered according to the signaling pathways altered by the different mutation. This type of functional data allows linkages to be made between key mutations thought to drive the tumor, and downstream pathways that may also be targeted through combination therapies. An open-access web portal, functional annotation of somatic mutations in cancer (FASMIC; http://bioinformatics.mdanderson.org/main/FASMIC) was also developed to ensure the data were broadly available [123]. These types of assays and the sharing of this function data will move us ever closer to the goal of precision oncology.

## 6. Conclusions

Mutations in different regions of p85α, even within the same domain, can exert very different cellular effects depending on which of the many p85α binding partners that they impact [11]. The fact that different surfaces of the p85α BH domain are involved in regulating PTEN and Rab5 (Figure 5; [105,113]) means that clinically, cancers which have deregulated Rab5 functions as a result of p85α mutations may require very different treatments as compared to those where PTEN is deregulated. Thus, not only is it important to determine the specific proteins that are mutated in a given cancer, but it will also be critical to determine the location, nature and impact of that specific mutation to define the most effective treatment for a particular patient.

## Figures and Tables

**Figure 1 biomolecules-09-00029-f001:**
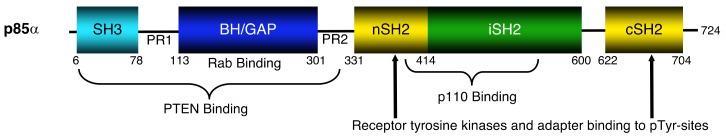
Domain structure of the p85α protein showing key regions involved in binding to several proteins. SH3 = Src homology region 3 domain, PR1 = proline-rich region 1, BH/GAP = breakpoint cluster homology or GTPase activating protein domain, PR2 = proline-rich region 2, nSH2 = N-terminal Src homology region 2 domain, iSH2 = interSH2 domain and cSH2 = C-terminal Src homology region 2 domain; pTyr = phosphotyrosine; PTEN = phosphatase and tensin homologue deleted on chromosome 10.

**Figure 2 biomolecules-09-00029-f002:**
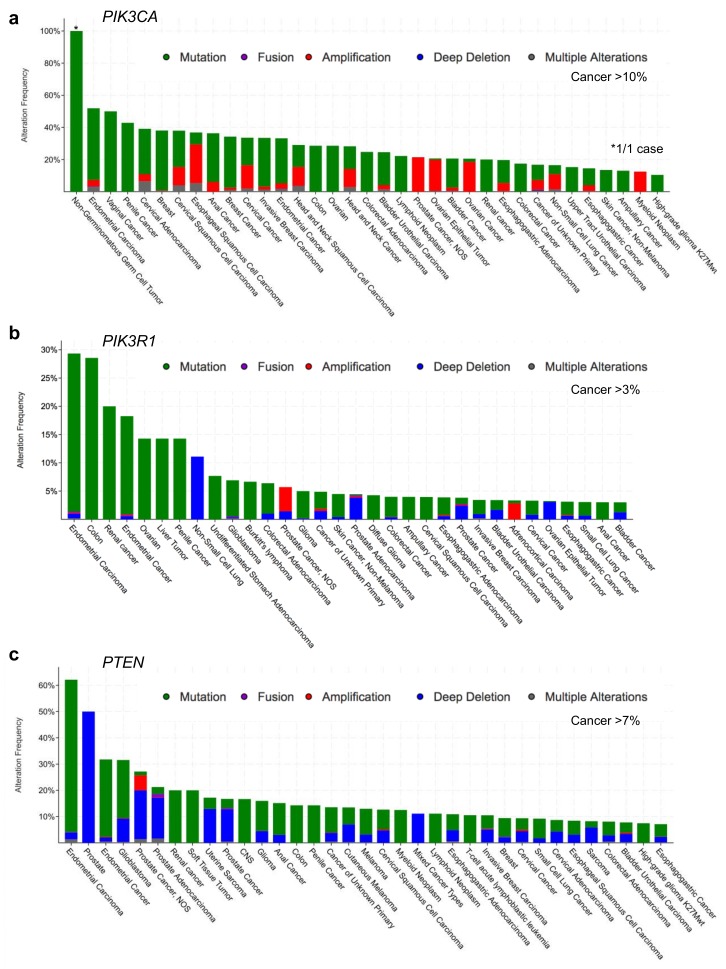
Human cancer types with the highest frequencies of alterations in (**a**) *PIK3CA*, (**b**) *PIK3R1*, and (**c**) *PTEN*. Cancer-associated alterations include: mutations (green), fusions (purple), amplifications (red), deep deletions (blue) and multiple alterations (grey). Figures were generated using cBioPortal [30,31]. NOS = not otherwise specified.

**Figure 3 biomolecules-09-00029-f003:**
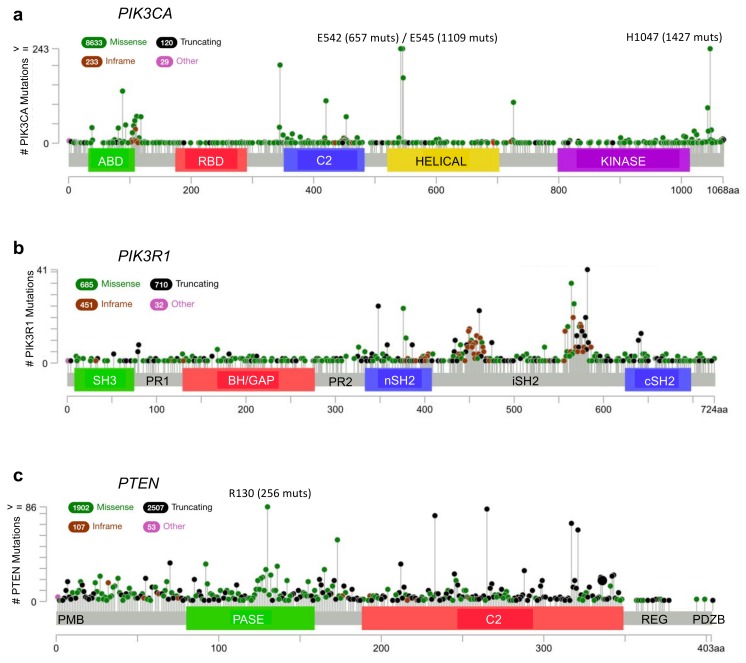
Frequency and distribution of cancer-associated mutations in key phosphatidylinositol 3-kinase (PI3K) pathway genes. (**a**) *PIK3CA*, (**b**) *PIK3R1*, and (**c**) *PTEN.* Missense (green), truncating (black), inframe (red) and other (magenta) are indicated and the frequency of each is reflected by the height of the line, according to scale indicated on the left. Hotspot mutations with very high frequencies beyond the scale are written above with the frequencies in brackets. *PIK3CA* encodes the p110α protein which contains five domains: an adapter binding domain (ABD), a Ras-binding domain (RBD), a C2 domain, a helical domain, and a lipid kinase domain. *PIK3R1* encodes p85α which includes an: SH3 domain, proline-rich region 1 (PR1), BH/GAP domain, proline-rich region 2 (PR2), and two SH2 domains (nSH2 and cSH2) on either side of an inter-SH2 domain (iSH2). The *PTEN* gene encodes the PTEN protein which has a plasma membrane binding domain (PMB), a dual specificity lipid/protein phosphatase domain (PASE), a C2 domain, a Ser/Thr-rich regulatory domain (REG), and a PDZ binding domain (PDZB). Figures were generated using cBioPortal [30,31].

**Figure 4 biomolecules-09-00029-f004:**
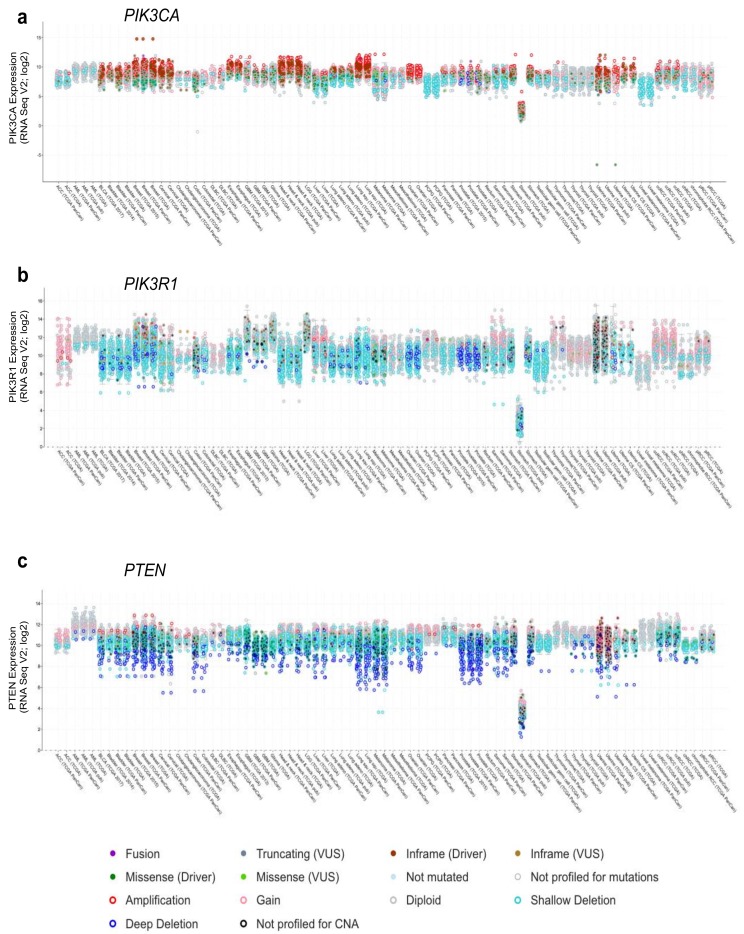
Gene expression across cancer types for (**a**) *PIK3CA*, (**b**) *PIK3R1*, and (**c**) *PTEN.* Gene expression (RNAseq, log2), together with genetic alteration information, is included by color-coding the samples as indicated, where available, e.g., missense driver mutations (dark green fill), missense variants of unknown significance (VUS; light green fill), not mutated (light blue fill), amplifications (red open), gain (pink open), deep deletion (dark blue open), and shallow deletion (cyan open). Figures were generated using cBioPortal [30,31].

**Figure 5 biomolecules-09-00029-f005:**
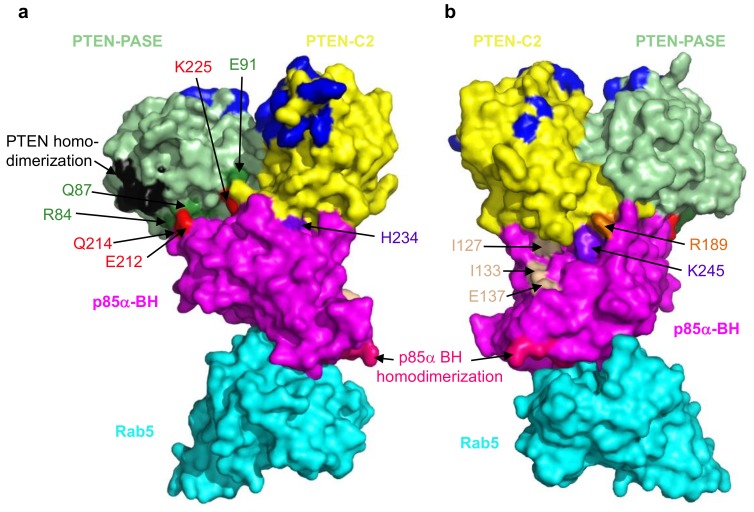
A model with two views showing some of the contacts between the PTEN and the p85α BH domain. (**a**,**b**) The PTEN-PASE domain (light green) and the p85α BH domain (magenta) interactions are shown R84:Q214, Q87:E212 and E91:K225 (interacting residues; PTEN-PASE are dark green: p85α are red). PTEN-C2 domain (yellow) and the p85α BH domain (magenta) interaction is shown, R189 (PTEN-C2 residue is orange): K245 (p85α BH domain residue is purple). There are additional contacts buried more deeply within the interface that are not visible in the space filling model. The PTEN homodimerization site (black) and PTEN membrane lipid binding residues (dark blue) are also shown. The p85α BH homodimerization residues (hot pink) and the three residues previously shown contribute to PTEN binding (I127, I133 and E137; tan) are also indicated. Rab5 (cyan) binds to a distinct region of the p85α BH domain that includes p85α residues L191 and V263.

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
