# Peer review of "Impact of p85α Alterations in Cancer"

_biomolecules, 2019, doi:10.3390/biom9010029_

Reviewer 1 Report

The manuscript by Anderson and coworkers about the p85a alterations in cancer describes p85a mutation, alterations in expression levels and its relationship to binding partners. They focus on alteration on the nterminal pof p85a since they regulate rho-family gtpases, pten.

To extend the scholarly contributions of the review we suggests that the authors make the following corrections and modifications:

1-In light of the content the authors should include a couple of lines that described the known phosphorylation of p85a (S43, S154, S208, T603 and T608 (http://www.jbc.org/content/292/33/13541.long)) as they have done with PTEN.

line 106, they should add a reference for the structural determination of the cancer-associated H1047R mutation

https://www.ncbi.nlm.nih.gov/pubmed/19805105

They should also mentioned that the H1047R seems to increase the attachment to the membrane.

Line 117, they should add a reference for the structural analysis of mutations such as g376R and K379N/E 

https://www.ncbi.nlm.nih.gov/pubmed/19962457

2- use of gene nomenclature to be consistent. For example genes should be capitalized and italicized like in the …., . So in line 30 and 31 the PIK3R1 and PIK3R3 should be corrected and capitalized accordingly. For example line 102 is correctly capitalized and italicized. Verify the preferences with the journal.

3-at the beginning, line 38, the authors should refer to PI3K class 1a and not just PI3K. The former more specific and narrow is correct. Leaving it vague is prone for misunderstanding since for instance PI3Kgama is activated by GPCR. The definition here would help throughout the manuscript

4-at the nomenclature level, it is recommended that  PI3,4,5P3  is called PI(3,4,5)P3. For example line 53, 231

5-line 65, replace “…protein is thought to be sequestered by” with “ protein is bound to p110a with a strong interaction and is also required for p110a stability.

Also p110b and p110d also bind p85  so more elaboration on the text is required.

6-line 76-77, sentence “Various drugs targeting PI3K, and other kinases within the pathway that are often constitutively activated in cancer cells, have been developed to treat dysregulation of the PI3K pathway.”

The part in between commas is too long and makes hard to read. Rewrite or make it 2 sentences

7- in the figure the labels PIK3CA and PIK3R1 change the font, they are hard to read

8- in the caption and about paragraph around 296-3007 the authors should comments on the possible effect of phosphorylation of Ser208 of p85 with the modeling of rab5 interaction

9- VUS, variants of unknown significance is used only in line 157 and  in line 311 so for clarity skip the acronym.

Other typos

Line 134 fibres (  

Author Response

The manuscript by Anderson and coworkers about the p85a alterations in cancer describes p85a mutation, alterations in expression levels and its relationship to binding partners. They focus on alteration on the nterminal pof p85a since they regulate rho-family gtpases, pten.

To extend the scholarly contributions of the review we suggests that the authors make the following corrections and modifications:

1-In light of the content the authors should include a couple of lines that described the known phosphorylation of p85a (S43, S154, S208, T603 and T608 (http://www.jbc.org/content/292/33/13541.long) PMID:28676499) as they have done with PTEN.

Response: We thank the reviewer for this suggestion. PTEN phosphorylation was discussed because it has an established functional role in the regulation of PTEN activity. In contrast, the role of even the best characterized p85a phosphorylation, S608, is unclear. Thus, given the already lengthy nature of this review, we have chosen to not include this information.

line 106, they should add a reference for the structural determination of the cancer-associated H1047R mutation

https://www.ncbi.nlm.nih.gov/pubmed/19805105

They should also mentioned that the H1047R seems to increase the attachment to the membrane.

Response: We have incorporated these suggestions, now line 120.

Line 117, they should add a reference for the structural analysis of mutations such as g376R and K379N/E 

https://www.ncbi.nlm.nih.gov/pubmed/19962457

Response: On line 117 (now line 131), we simply indicate the most frequent p85a mutations, whereas on line 123 we mention the impact of mutation of K379, so we have added this reference to line 123 instead.

2- use of gene nomenclature to be consistent. For example genes should be capitalized and italicized like in the …., . So in line 30 and 31 the PIK3R1 and PIK3R3 should be corrected and capitalized accordingly. For example line 102 is correctly capitalized and italicized. Verify the preferences with the journal.

Response: Corrected for consistency, line 30.

3-at the beginning, line 38, the authors should refer to PI3K class 1a and not just PI3K. The former more specific and narrow is correct. Leaving it vague is prone for misunderstanding since for instance PI3Kgama is activated by GPCR. The definition here would help throughout the manuscript

Response: Added, line 38.

4-at the nomenclature level, it is recommended that  PI3,4,5P3  is called PI(3,4,5)P3. For example line 53, 231

Response: Changed throughout manuscript on lines 48, 56, 67, 177, 178 and 179. Also changes PI4,5P2 to PI(4,5)P2 for consistency on lines 47 and 67.

5-line 65, replace “…protein is thought to be sequestered by” with “ protein is bound to p110a with a strong interaction and is also required for p110a stability.

Response: Changed, now on line 68.

Also p110b and p110d also bind p85  so more elaboration on the text is required.

Response: These are mentioned very briefly on line 111 and are not the focus of this review.

6-line 76-77, sentence “Various drugs targeting PI3K, and other kinases within the pathway that are often constitutively activated in cancer cells, have been developed to treat dysregulation of the PI3K pathway.”

 The part in between commas is too long and makes hard to read. Rewrite or make it 2 sentences

Response: Have changed to: “Various drugs targeting PI3K and other constitutively activated kinases within the pathway have been developed to treat dysregulation of the PI3K pathway” now on lines 78 and 79.

7- in the figure the labels PIK3CA and PIK3R1 change the font, they are hard to read

Response: We have used a standard Helvetica 10-point italics for these gene names for all the figure labels.

8- in the caption and about paragraph around 296-3007 the authors should comments on the possible effect of phosphorylation of Ser208 of p85 with the modeling of rab5 interaction

Response: p85a S208 is not particularly close to either PTEN or Rab5 in the model shown in Figure 5. This residue is located on the magenta p85a BH domain below the red E212 and oriented towards the front left surface of the domain. Therefore, we don’t think that it is necessary to speculate about the possible impacts of phosphorylation at this site.

9- VUS, variants of unknown significance is used only in line 157 and  in line 311 so for clarity skip the acronym.

Response: The figure was generated using cBioPortal and as such the use of the abbreviation VUS appears in 3 instances within the figure codes generated by cBioPortal. We therefore believe that it is important to define the acronym within the figure 4 legend (line 172). The use of the same acronym in line 337 simply expands upon this class of mutants and provides needed continuity. For these reasons, we have kept the abbreviation.

Other typos

Line 134 fibres (  

Response: Corrected typo, now line 148.

Reviewer 2 Report

The review by Anderson and colleagues presents a one-sided view of the literature with regard to the GAP activity of p85. Other groups have presented data contradicting these findings. The review should acknowledge the controversy and cite these papers:

Ahmed et al. JBC 1994 269:17642-17648 shows that residues crucial for GAP activity are not present in p85.

Amin et al. JBC 2016 20353-20371 presents a rigorous analysis of Rho GTPases GAPS, and concludes that p85 does not possess GAP activity.

Christoforidis et al. Nature Cell Biol. 1999 1:249-252 uses IVT to show that Rab5(GTP) binds to p110beta and p85alpha/p110beta dimers, but not to p85alpha alone.

Author Response

The review by Anderson and colleagues presents a one-sided view of the literature with regard to the GAP activity of p85. Other groups have presented data contradicting these findings. The review should acknowledge the controversy and cite these papers:

Ahmed et al. JBC 1994 269:17642-17648 shows that residues crucial for GAP activity are not present in p85.

Response: This paper uses primary amino acid sequence alignments to suggest that the BH domain of p85a is quite different in sequence as compared to several other BCR-family GAPs. The more recent structural data for the p85a BH domains (Musacchio et al., PNAS 1996 93: 14373-14378; Mellor et al., Sci Rep 2018 8:7108), and modeled structural interactions between the p85a BH domain and Rab5 (Cheung et al., Elife 2015 4: e06866; Mellor et al., Sci Rep 2018 8:7108; Marshall et al., Oncotarget 2018 9: 36975-36992), coupled with an enzymatic GAP assay argue that the critical catalytic residues are present within the p85a BH domain.

Other labs have confirmed that p85a has Rab5 GAP activity both in vitro (Dou et al., Mol Cell, 2013; PMID: 23434372) and in vivo (Diaz et al., J Cell Sci, 2014 127:2401-2406). Further, the structural alignment of the p85a BH domain – Rab5 with Cdc42GAP – Cdc42 (Figure 6 of Mellor et al., Sci Rep 2018 8:7108) shows that the structure of the BH/GAP domain of p85a overlays well with that of Cdc42GAP and positions R151 to function as the catalytic arginine finger for p85a. We have also previously shown that mutation of R151 of p85a compromises its Rab5 GAP activity (Chamberlain et al., JBC, 2004 279: 48607-48614).

Since the Ahmed paper is more than 20 years old and tries to make conclusions based on amino acid sequence alignments, we don’t feel that it’s warranted to mention it.

Amin et al. JBC 2016 20353-20371 presents a rigorous analysis of Rho GTPases GAPS, and concludes that p85 does not possess GAP activity.

Response: The Amin paper did not claim to test p85a as a GAP for Rab5, since its focus was on Rho GTPases. Most importantly, this paper also used a different assay system to measure GAP activity (stopped flow measurements) over the course of 2 seconds, rather than the radionucleotide-loaded assay system that we and others (Dou et al., Mol Cell, 2013 50: 29-42) use with measurements taken after 20-30 minutes. We have always suggested that p85a has weak GAP activity, so that after only 2 seconds in the assay, it’s not surprising that little or no activity was detected. We have updated the text in the paper to indicate that the BH domain possesses weak GAP activity and have added in this citation (see line 195).

Christoforidis et al. Nature Cell Biol. 1999 1:249-252 uses IVT to show that Rab5(GTP) binds to p110beta and p85alpha/p110beta dimers, but not to p85alpha alone.

Response: Subsequent studies have shown that the p110beta – p85alpha complex can compete with p85alpha for binding to Rab5-GTP to block the Rab5 GAP activity of p85alpha (Dou et al., Mol Cell, 2013 50: 29-42). This work further suggested that the binding affinity of Rab5 is higher for p110beta than for p85alpha, consistent with the results of the Christoforidis et al. paper.

Reviewer 3 Report

The authors present a comprehensive and detailed review of p85 alternations and their potential impact on known protein-protein interactions. Specifically, the review details the interaction of p85 with  PTEN and Rho-GTPases, and how mutations and/or changes in p85 expression might affect the activity of these proteins. A few additions addressed below would benefit the review.

The authors describe the interaction between CDC42/Rac and p85 and how this may increase PI3K signaling. More information should be added to more fully address the potential positive feedback loop this can create due to activation of PIP3 regulated GEFs at the cell membrane.

It is not entirely clear in the review if there is evidence human tumors that reduced p85 expression/mutations that disrupt homodimerization directly correlate with loss of PTEN protein expression. This should addressed. Also, are there cancer-associated mutations in PTEN that are known to disrupt binding to p85?

p85 is also known to interact with IRS1, sequestering it away from P110a and inhibiting PI3K signaling during starvation conditions. This would be a nice addition to the introduction.

Author Response

The authors present a comprehensive and detailed review of p85 alternations and their potential impact on known protein-protein interactions. Specifically, the review details the interaction of p85 with  PTEN and Rho-GTPases, and how mutations and/or changes in p85 expression might affect the activity of these proteins. A few additions addressed below would benefit the review.

The authors describe the interaction between CDC42/Rac and p85 and how this may increase PI3K signaling. More information should be added to more fully address the potential positive feedback loop this can create due to activation of PIP3 regulated GEFs at the cell membrane.

Response: We have added the following additional information: ” including Vav1 and Sos1, both of which require PI3,4,5P3 for their activity [52,53]. This creates a positive feedback loop due to activation of PI3,4,5P3 regulated GEFs at the cell membrane.” To lines 177-179.

It is not entirely clear in the review if there is evidence human tumors that reduced p85 expression/mutations that disrupt homodimerization directly correlate with loss of PTEN protein expression. This should addressed. Also, are there cancer-associated mutations in PTEN that are known to disrupt binding to p85?

Response: added in new information: “They also showed a corresponding decrease in PTEN levels, consistent with the loss of p85a-mediated PTEN stabilization.” On lines 319 -320.

“Based on this model, there are cancer-associated mutations in PTEN that would be predicted to disrupt binding to p85a, including: D84G, Q87P, Y88C/H/N/S, E91Q, E99K, R189K, K342N and K344R (Marshall, 2018 ref). Similarly, there are cancer-associated mutations in p85a that would be predicted to disrupt binding to PTEN, including: Q214L, W237L and K245T, according to the COSMIC and cBioPortal databases. (added into lines 300-304).

p85 is also known to interact with IRS1, sequestering it away from P110a and inhibiting PI3K signaling during starvation conditions. This would be a nice addition to the introduction.

Response: While we agree that this would make a nice addition to the introduction, the manuscript is already quite long and so we have not added this further information.